

# Commercial arbuscular mycorrhizal fungal inoculant failed to establish in a vineyard despite priority advantage

Corrina Thomsen[1], Laura Loverock[1], Vasilis Kokkoris[2,4], Taylor Holland[1], Patricia A. Bowen[3] and Miranda Hart[1]

[1] Department of Biology, University of British Columbia Okanagan, Kelowna, BC, Canada
[2] Department of Biology, University of Ottawa, Ottawa, ON, Canada
[3] Summerland Research and Development Centre, Agriculture and Agri-food Canada, Summerland, BC, Canada
[4] Ottawa Research and Development Centre, Agriculture and Agri-Food Canada, Ottawa, ON, Canada

## ABSTRACT

**Background**. Arbuscular mycorrhizal (AM) fungi associate with most plants and can increase nutrient uptake. As a result, commercial inoculants called "biofertilizers" containing AM fungi have been developed and marketed to increase plant performance. However, successful establishment of these inoculants remains a challenge, and may be negatively impacted by competition with fungi already present (priority effects). Perennial agriculture may be more amenable if inoculants can be successfully established on crops prior to field planting.

**Methods**. Here, we inoculate grapevine (*Vitis vinifera*) with a commercial inoculant in three treatments designed to manipulate the strength and direction of priority effects and quantified the abundance of the fungal strain before and after introduction using droplet digital PCR (ddPCR).

**Results**. We found that the introduced strain did not establish in any treatment, even with priority advantage, and inoculated vines did not differ in performance from non-inoculated vines. Fungal abundance was not greater than in pre-inoculation soil samples during any of the five years sampled and may have been impaired by high available phosphorus levels in the soil. This study highlights the need to understand and evaluate how the management of the agricultural system will affect establishment before introduction of "biofertilizers", which is often unpredictable.

## INTRODUCTION

Currently, there is a need to improve agricultural phosphorus use efficiency and sustainability (*Cordell & White, 2013*), which could be accomplished through optimizing microbially mediated nutrient cycling to promote maximum crop uptake (*Vance, 2001*; *Hamel & Strullu, 2006*; *Thirkell et al., 2017*). In response, inoculants called "biofertilizers" have been developed which aim to improve plant growth through the addition of beneficial

Corresponding authors
Corrina Thomsen,
corrina.thomsen@ubc.ca
Miranda Hart, miranda.hart@ubc.ca

soil microbes, of which arbuscular mycorrhizal (AM) fungi are commonly used (*Thirkell et al., 2017*).

AM fungi are obligate root symbionts that associate with the vast majority of terrestrial plants, and transfer nutrients, primarily phosphorus, to their host plant in exchange for carbon (*Smith & Read, 2008*). Mycorrhizal plants rely on these fungal partners for nutrients, primarily nitrogen and phosphorus, and, among other benefits, they can provide a significant percentage of a plant's phosphorus requirements, as well as potentially access phosphorus that the host cannot access (*Smith & Read, 2008*). It is possible that optimizing the interactions between AM fungi and crops, through either increasing total percent colonization or the relative abundance of more mutualistic partners, could increase crop nutrient uptake. Inoculation with a commercial "biofertilizer" could do so if the abundance of AM fungi in the field is low and limiting the total root colonization of the crops, or if the commercial isolate (strain) is superior to the native isolates and disproportionately increases plant growth. If this could be accomplished, it could reduce the amount of fertilizer required, and could allow crops to access reserves of "legacy" phosphorus leftover in the soil from past fertilization which is not easily accessed by plant roots (*Condron et al., 2013*; *Cordell & White, 2013*).

## Challenges for practical applications of AM fungal biofertilizers

At last estimate, the market for these inoculants had grown to include over thirty producers worldwide, and they are available to home gardeners as well as large-scale growers. Despite their use, there are many unanswered questions regarding their efficacy and best practices. In some cases, these inoculants have been successful at increasing plant growth and/or yield (*Pellegrino et al., 2011*; *Pellegrino, Öpik & Bonari, 2015*; *Hernádi et al., 2012*). However, in other cases, fungal inoculation did not affect plant performance (*Ortas et al., 2011*; *Janoušková et al., 2013*; *Emam, 2016*). While the cause of these failures is not clear, field applications are typically less successful than in greenhouse applications (*Lekberg & Koide, 2005*).

The majority of studies on the effect of commercial AM fungal inoculants on plant growth and/or yield have been conducted in greenhouse (*Berruti et al., 2016*), but this approach does not capture the complexities of field conditions. Often, these studies are conducted in the absence of or with a simplified AM fungal community. Such interactions could potentially inhibit the establishment and/or performance of the inoculant through priority effects (*Verbruggen et al., 2013*). *Priority effects* occur when the first species to colonize an area (in this case, host roots) experiences an advantage and ultimately affects the composition of the assembled community. This effect has been observed in many taxa (*Alford & Wilbur, 1985*; *Ladd & Facelli, 2008*; *Fukami et al., 2010*; *Peay, Belisle & Fukami, 2012*) including AM fungi (*Werner & Kiers, 2015*). While many field collected roots will show a diversity of AM fungi (*Vandenkoornhuyse et al., 2002*), it is possible that subsequent establishment of introduced AM fungi may become progressively less likely as host resources become increasingly monopolized. This phenomenon presents a direct challenge for biofertilizers, which must compete with fungi already present in the field.

## Quantifying biofertilizer establishment

Establishment and persistence of commercial AM fungal inoculants is crucial for successful application of the biofertilizer; if the inoculant does not survive, it is unlikely to result in any benefit. However, even under the most benign conditions in sterile greenhouse pots, establishment failed in 75% of the commercial inoculants tested (*Faye et al., 2013*). Field establishment may be even less successful for the reasons above, and reduced establishment could contribute to the reduced benefit observed relative to greenhouse studies. In a meta-analysis, *Lekberg & Koide (2005)* showed a lower average mycorrhizal response in field studies compared to studies conducted in greenhouse. However, few studies have used specific molecular techniques to monitor the establishment and persistence of a single commercial inoculation event for two or more years.

It is possible that improved inoculation practices may be able to increase the likelihood of successful establishment. With perennial crops, such as fruit trees and grapevines, young plants can be "pre-inoculated" in the greenhouse before being transferred to the field. By establishing the commercial inoculant in the absence of a native AM fungal community before transplanting, the priority advantage could be given to the commercial inoculant. In this experiment, we tested three inoculation strategies on grapevine that differ in the strength and direction of priority effects. We monitored the vines over five years using isolate-specific molecular techniques to determine the abundance of the introduced commercial isolate and recorded the response of economically important measures of vine performance. To the best of our knowledge, this study represents the longest study tracking inoculum persistence in the field.

# MATERIALS & METHODS

## Study site and vine characteristics

This experiment was conducted at Kalala Organic Estate Winery located at 49.84 N, 119.64 W in West Kelowna, British Columbia, Canada (photos of the site are included in Supplementary Materials). West Kelowna is located in the Ponderosa Pine Very Dry Hot Biogeoclimatic Zone (*Ministry of Forests, Lands, Natural Resource Operations and Rural Development, 2018*), with an annual average temperature of 14.7 °C and total precipitation of 344.5 mm (*Environment Canada, 2018*). The soil is primarily sandy loam with rapid drainage and 42% coarse fragment material (*Wittneben, 1986*). This site has been under the same owners for the last 15 years and was not inoculated with AM fungi during that time.

All vines used were Pinot Noir varietals grafted onto 3309C rootstocks purchased from Sunridge Nurseries (Bakersfield, CA). Experimental vines were established in 2013 in a section of the vineyard approximately 50 m × 200 m and planted within empty positions in the vineyard among non-experimental Pinot Noir vines. After planting, vines were maintained according to standard organic practice by the vineyard.

## Experimental factors and timing strategies

For this experiment, a factorial design was established with two factors, inoculation and timing strategy, and vines from each treatment were randomly distributed among the

**Table 1** **Samples sizes of each of the inoculation strategies used in analyses of inoculant establishment and persistence.** Plants were either inoculated prior to planting (Pre), inoculate at planting (Co.) or inoculated onto an existing vine (Est.). Equal numbers of vines in each category were inoculated with a live fungus (Inoc.) or inoculated with an inert carrier (Cont.).

| Sampling Period | Pre. | | Co. | | Est. | | Total |
|---|---|---|---|---|---|---|---|
| | Inoc. | Cont. | Inoc. | Cont. | Inoc. | Cont. | |
| May 2013 | 0 | 0 | 7 | 9 | 14 | 8 | 38 |
| October 2013 | 8 | 8 | 7 | 7 | 13 | 7 | 50 |
| October 2014 | 7 | 8 | 7 | 9 | 14 | 8 | 53 |
| October 2015 | 7 | 8 | 7 | 5 | 14 | 8 | 49 |
| October 2016 | 8 | 8 | 7 | 8 | 13 | 7 | 51 |
| October 2017 | 7 | 8 | 7 | 9 | 14 | 8 | 53 |

available planting locations in the vineyard. Within each timing strategy, half of the vines were inoculated with MYKE Pro Greenhouse G granular inoculum (Premier Tech Ltd., Rivière-du-Loup, Quebec, Canada) containing carrier (peat moss and perlite), and hyphal fragments and spores of *Rhizoglomus irregulare* DAOM 197198 (~20 propagules/g), and the remaining half were non-inoculated and exposed to sterile carrier only. Although this product is labelled "Greenhouse", the company states that it can also be used when transplanting into soil and is suitable for perennial plants. The company information still states that the product contains *Glomus intraradices* DAOM 197198, however this isolate was reclassified under *Glomus irregulare* (*Stockinger, Walker & Schüzler, 2009*), then the species was renamed *Rhizophagus irregularis* (*Schüzler & Walker, 2010*; *Formey et al., 2012*), and most recently *Rhizoglomus irregulare* (*Sieverding et al., 2014*).

Three timing strategies were used as follows:

1. Pre-inoculation (Pre): Grafted new vines were inoculated on May 24th 2013 with 60 mL of MYKE Pro Greenhouse G inoculum and grown in a greenhouse in a 2-gallon pot containing sterile Sunshine Mix #4 soil (SunGro Horticulture, Bellevue, WA). Vines were watered daily and grown in a randomized arrangement at 16-hour days and 25 °C for 4 weeks before transplanting into the field with trimmed roots.

2. Co-inoculation (Co): Grafted new vines were planted between May 30th and June 4th 2013 with 60 mL of MYKE Pro Greenhouse G inoculum in the hole at the time of planting. Roots were trimmed and planted vines received 3L water after planting.

3. Established (Est): One-year-old vines growing in the vineyard were randomly selected and inoculated in May 2013 by taking 3 soil cores 10 cm from the vine (left, right, and centre). 20 mL of MYKE Pro Greenhouse G inoculum were added to each hole for a total of 60 mL and watered well.

A total of 120 vines were initially established in the vineyard, and the final sample size used in each test is reported in Tables 1 and 2. All plants had their roots trimmed at the time of planting.

## Soil chemistry

Three soil cores were taken at each experimental vine in October 2013 and pooled, dried at 60 °C for 24 h, and sieved in a 2 mm sieve. Sieves were sterilized by thoroughly flaming

**Table 2 Summary of the number of observations from each inoculation strategy used in analyses of grapevine performance.** Note that for tests that include multiple years (i.e., 2013–2015 for shoot length), total $n$ pooled from all years is presented.

| Test | Pre. Inoc. | Pre. Cont. | Co. Inoc. | Co. Cont. | Est. Inoc. | Est. Cont. | Years | Total |
|---|---|---|---|---|---|---|---|---|
| Survival | 17 | 16 | 18 | 17 | 20 | 16 | 1 | 104 |
| Shoot length | 45 | 47 | 53 | 49 | 42 | 34 | 3 | 270 |
| Shoot diameter | 29 | 34 | 36 | 32 | 34 | 27 | 2 | 192 |
| Cluster production | 14 | 15 | 18 | 14 | 14 | 13 | 1 | 88 |
| Cluster number | 29 | 34 | 36 | 32 | 34 | 27 | 2 | 192 |

between samples. Each of the 52 samples were analyzed for pH, phosphorus, and nitrogen by Zenalytics Laboratories in Kelowna, BC. Soil pH in $H_2O$ was tested using a pH meter, and total inorganic and organic nitrogen was tested with the Kjeldahl method (*Rutherford et al., 2007*). Phosphorus was extracted using the Mehlich III extractant (*Mehlich, 1984*) and tested with the ICP-MS method, which primarily detects orthophosphate (*Cade-Menun et al., 2018*).

## Fungal characteristics in the soil

Soil from cores, collected and processed as described above, and fresh roots collected from vines in October 2017 were sent to the University of California Riverside for analysis. Soil hyphal length was measured following *Miller, Jastrow & Reinhardt (1995)*, and soil spore density was determined following *Klironomos et al. (1993)*. Percent root colonization by hyphae, arbuscules, and vesicles was determined using the McGonigle method (*McGonigle et al., 1990*). Samples were stored at −20 °C and shipped frozen.

## Sampling and DNA extraction

To determine establishment of the commercial isolate on experimental vines, soil cores were from vines in May 2013 prior to inoculation, and October 2013-2017. Three cores were taken from each vine, 10 cm away from the trunk left, right, and centre. The soil was pooled, dried at 60 °C for 24 h, homogenized, and sifted in a sterile 2 mm sieve. Sieves were thoroughly flame-sterilized between each sample. Processed soil was stored at −20 °C until DNA extraction.

Establishment of the isolate on off-target vegetation was determined by taking 199 total root samples in August 2017 from 10 common species found in the interrow vegetation and which represented a range of rooting and life history strategies (Table S1). 50 cm × 50 cm quadrats were placed in front of each experimental vine, and if any of the 10 focal species were present, one representative of each species was randomly selected and sampled by harvesting the plant and removing a portion of the root system. A single soil core was also taken from the center of each of 102 quadrats. To assess the spread from the inoculated block, soil was collected and extracted in July 2017 using the same protocol from three soil cores taken from 80 random locations in interrows north and east of the inoculated block.

All DNA was extracted using the PowerSoils DNA kit (MoBio, Inc., acquired by Qiagen in 2017), and stored at −80 °C, or −20 °C when in frequent use. Root samples were frozen

in liquid nitrogen and pulverized by mechanically shaking with ceramic beads prior to extraction.

## Droplet digital PCR (ddPCR) analysis

DNA extracted from soil and root samples was tested for the presence of *R. irregulare* DAOM 197198 using a probe-based ddPCR assay on the Bio-Rad QX100 Droplet Digital PCR System (Bio-Rad Laboratories, Inc., Hercules, California). This technique allows accurate detection and quantification of target DNA by partitioning the sample into >10,000 droplets prior to the PCR reaction. Primers and probe for this assay were designed to target the *cox3-rnl* intergenic region (*Kokkoris et al., 2019*), and the sequences are as follows (see Table S2 for further details):

Forward: 5′-AGCAAATCTAAGTTCCTCAGAG-3′
Reverse: 5′-ACTTCTATGGCTTTGTACAGG-3′
Probe: 5′-FAM/CCCACCAGG/ZEN/GCAGATTAATCTTCCTT/3IABKFQ-3′

These primers amplified only DAOM 197198 of 23 *R. irregulare* strains and 12 AM fungal species tested and have no predicted specificity to other AM fungal isolates. They also did not amplify in environmental soil samples taken from natural areas in nearby Merritt, BC, that should contain a similar background AM fungal community with a minimal risk of agriculturally introduced AM fungi.

The reaction contains 10 μl Bio-Rad ddPCR Supermix for Probes, 2 μl of Integrated DNA Technologies PrimeTime qPCR Assay mixture with a final concentration of 500 nM each forward and reverse primer and 250 nM probe, 1 μl of template (extracted DNA from soil), and 7 μl of dH20 for a total of 20 μl. Positive controls contained amplicon of the target sequence which was produced from DNA extracted from onion roots colonized by the biofertilizer used, and negative controls received sterile water. Three negative control wells were included on each plate, and all three wells must have zero positive droplets for the results of the plate to be considered valid. The total volume was transferred to the Bio-Rad QX100 Droplet Generator with 70 μl of Bio-Rad Droplet Generator Oil for Probes to produce 40 μl of droplets. We required a minimum of 10,000 droplets in an individual well during measurement for the well to be considered valid. The droplets were cycled at: 95 °C for 10 min, 50 cycles of 94 °C for 30 s and 57.8 °C for 2 min, 98 °C for 10 min, and 4 °C hold. After cycling, the amount of fluorescence in the droplets was measured by the Bio-Rad QX100 Droplet Reader. The number of positive (fluorescent) droplets in the sample is converted to the estimated number of target copies per μl of template using QuantaSoft (Version 1.7.4.0917, Bio-Rad Laboratories Inc.). The threshold for a positive call was manually set just below the bottom of the cloud produced in the positive control well on each plate (see Fig. S1). A high threshold was chosen because droplets that do not reach the fluorescence of this level are more likely false positives, as droplets that contain at least one copy of target sequence should reach roughly the same final fluorescence. Further, the primer set alone does amplify non-target sequence from the site, and although the probe-binding region is not a good match on these non-target amplicons, it may bind occasionally and produce some fluorescence (Fig. S2). However, if this does occur, the

fluorescence should be much lower than true positives and thus excluded by our high threshold.

## GPS coordinates and distance calculations

GPS coordinates for each of the samples taken outside the inoculated block, as well as the corners of the inoculated block, were taken using the app "GPS Status" version 1.2 on an iPhone 6S. Each recording was taken after the location accuracy had stabilized to within 5 m. The distance between the sample site and the nearest edge of the inoculated block was calculated in Microsoft Excel version 14.7.2 using the Haversine formula, as follows:

$$Distance = ACOS(COS(RADIANS(90 - Lat_1)) \times COS(RADIANS(90 - Lat_2))$$
$$+ SIN(RADIANS(90 - Lat_1)) \times SIN(RADIANS(90 - Lat_2))$$
$$\times COS(RADIANS(Long_1 - Long_2))) \times 6,371,000.$$

This formula requires the latitude and longitude input to be in decimal degree format and provides the Great Circle Distance in metres.

## Vine measurements

For each vine, length of the longest new shoot (cm), shoot diameter (mm), and the number of grape clusters were recorded to assess vine response to the inoculant. The number of grape clusters was chosen to get a measure of vine yield, and the other variables measured were measured as they are used in determining vine vigor and balance. Vine survival was determined by visiting all vines on May 31st, 2017, well after vines had begun leafing out. Any vine that did not have any fresh growth, had been removed, or had been replaced was recorded as "dead". Only vines with new growth that season were recorded as "alive".

## Statistical methods

All statistical analyses were conducted using RStudio version 1.1.383 (*R Core Team, 2018*), and $\alpha = 0.05$ was used for all analyses. Sample sizes for each test are shown in Tables 1 and 2.

First, spatial autocorrelation of isolate abundance was determined using the function "Moran.I" from the package *ape* version 5.0 (*Paradis & Schliep, 2018*). A row-normalized matrix of inverse pair-wise distances between vines was used as the weight matrix, generated using "dist" from the package *stats* version 3.6.1 (*R Core Team, 2018*) and taking the inverse and normalizing. Converting the matrix to this format provides greater weight to nearby vines, as the influence of a neighbouring vine is inversely proportional to its distance and ensures the influence of neighbours add to one for each vine, so that the weights are expressed as proportions and thus comparable. The distance between rows in the vineyard is 3.05 m (10 ft) and the distance between vines is 1.22 m (4 ft). This scale was maintained in the distance matrix by multiplying the row numbers by ten and the within-row vine position by four. Establishment of the isolate was then tested by comparing whether inoculated vines were more likely to test positive for the isolate than non-inoculated vines using a Fisher's Exact test through the "fisher.test" function from the *stats* package, version 3.6.1 (*R Core Team, 2018*). Limited positive data prevented statistical analysis of effects of inoculation and strategy on abundance and spread from the inoculated block.

Differences in grapevine survival and the percent of vines producing fruit between inoculated and non-inoculated vines were tested using a Fisher's Exact test. A global Fisher's test was performed first on all of the data to test whether inoculation alone reduced mortality, and then the data were subset by strategy and the test was repeated to determine whether there was an interaction with strategy. Analysis of the percent of vines fruiting was restricted to 2015 because this was the most recent year for which we had complete data.

The effect of inoculation and the interaction with strategy on shoot length, diameter, and the number of grapes produced by fruiting vines was tested using a linear mixed-effects model (LMM) approach through the function "lmer" from the package *lme4* 1.1.21 (*Bates et al., 2015*). Inoculation and its interaction with strategy were coded as fixed factors and the year the measurement was taken was coded as a random factor. Log-transformations were used to normalize data as needed. The model was assessed using the function "Anova" from the package *car* version 3.0.5 (*Fox & Weisberg, 2019*).

# RESULTS

## Soil characteristics
### Chemical
The soil pH at the experimental site was neutral, with a median pH of 7.03 (Table S3). Phosphorus content as measured by Mehlich III/ICP-MS ranged from 184.7–548.6 mg/kg with a median of 320.4 mg/kg (Table S3). Median total nitrogen from the Kjeldahl method was 0.3%, ranging from 0-0.47% (Table S3).

### Fungal
Soil collected from the study site had a median hyphal length of 0.93 m/g of soil and a median of 18 spores/g of soil (Table S4). Root fragments from experimental vines showed a median colonization of 35.5% (Table S5), consisting of 5% arbuscules (range = 0–14%), 10% vesicles (range = 5–30%), and 25.5% hyphae (range = 10–39%).

## Establishment and spread of the introduced isolate
Overall, there was no evidence that the isolate established over the course of our experiment. The abundance of the isolate in the soil was very low, both in terms of the percentage of samples that were positive and the average abundance in each sample (Table 3). Importantly, the target isolate was also detected in 10.4% of samples taken prior to inoculation, indicating that the isolate was already present in the soil before our experiment began. In comparison, the percentage of positive soil samples post inoculation ranged from 3.8% to 20.4% and fluctuated between years rather than showing a consistent trend. Soil samples taken from vines that had been inoculated were not more likely to be positive for the isolate than vines that had not been inoculated, in any year (Fisher's Exact test, *P*: 0.28-1, Table 3).

The low number of positive soil samples precluded statistical analysis, but there were no apparent effects. Soil samples from Established vines were marginally more likely to be positive (2–4 positive vines per sampling period) than vines of the Pre- and Co-inoculation strategies (0–1 vines). However, two of the positive Established vines were positive before inoculation and continued to test positive in subsequent years.

**Table 3** **Number of samples positive for *R. irregulare* DAOM 197198, and the results of the Fisher's Exact test for the effect of inoculation on the likelihood of detecting the isolate.** Note that Pre-inoculated samples were not available from the May 2013 samp.

|  | May 2013 | October 2013 | October 2014 | October 2015 | October 2016 | October 2017 |
|---|---|---|---|---|---|---|
| Positive samples | 4 (10.5%) | 8 (16%) | 2 (3.8%) | 10 (20.4%) | 7 (13.7%) | 4 (7.5%) |
| Positive samples which were inoculated | 2 (50%) | 5 (62.5%) | 2 (100%) | 4 (40%) | 4 (57%) | 3 (75%) |
| *Pre-inoculated* | n/a | 0 | 0 | 1 | 0 | 1 |
| *Co-inoculated* | 0 | 1 | 0 | 0 | 1 | 0 |
| *Established* | 2 | 4 | 2 | 3 | 3 | 2 |
| Total samples | 38 | 50 | 53 | 49 | 51 | 53 |
| *P*-value (Fisher's Exact) | 1 | 1 | 0.49 | 0.28 | 1 | 0.62 |
| Median copy number | 9.8 | 13.0 | 3.8 | 6.2 | 6.6 | 5 |
| Mean copy number | 22.5 | 1939.7 | 3.8 | 9.7 | 7.6 | 5 |
| Range copy number | 2.6-68 | 1.6-13440 | 1.6-6 | 2.4-126 | 2.8-10.8 | 2.4-8.8 |

**Table 4** **ddPCR results from the positive samples of interrow soil and vegetation roots from within the inoculated block in August 2017.** These positive samples represented 2.9% of soil samples and 1.5% of root samples. Only 1.43 TP and 1.43 were taken from the same.

| Sample ID | Sample description | Input material weight (g) | Copies/uL | Copies/g input |
|---|---|---|---|---|
| 2.24 BC | *B. carinatus* | 0.08 | 1.4 | 1750 |
| 1.43 TP | *T. pratense* | 0.27 | 1.4 | 518.5 |
| 1.20 BM | *M. lupulina* | 0.1 | 1.4 | 1400 |
| 1.3 | soil | 0.25 | 1.8 | 720 |
| 1.43 | soil | 0.25 | 1.4 | 2560 |
| 2.15 | soil | 0.25 | 6.4 | 560 |

Low abundance of the target isolate was also reflected in interrow vegetation, where 3 of 199 root samples and 3 of 102 soil samples were positive (Table 4). Positive root samples came from roots of *Bromus carinatus*, *Trifolium pratense*, and *Medicago lupulina*, which represent both fibrous and taproot species as well as annual and perennial strategies.

All soil samples taken beyond the inoculated block to determine the spread of the isolate were negative.

## Lack of spatial auto-correlation

No significant spatial auto-correlation was detected during any sampling period (See Table S6 for statistical results). The correlation coefficients were small, ranging from 0.01 to -0.05, with *P*- values ranging from 0.27–0.82. The absence of spatial auto-correlation can also be seen in the lack of consistent, positive correlation coefficients between nearby vines in Fig. 1 and the absence of defined spatial structure in abundance in Figs. 2A–2C (See Figs. S3–S8 for all years).

## Grapevine survival

Twenty-seven of 104 vines had died or had been replaced by May 2017. Overall, non-inoculated vines experienced higher mortality (30.6%) than inoculated vines (21.8%), but this difference was not statistically significant (Fig. S9; Fisher's Exact, odds ratio = 1.574,
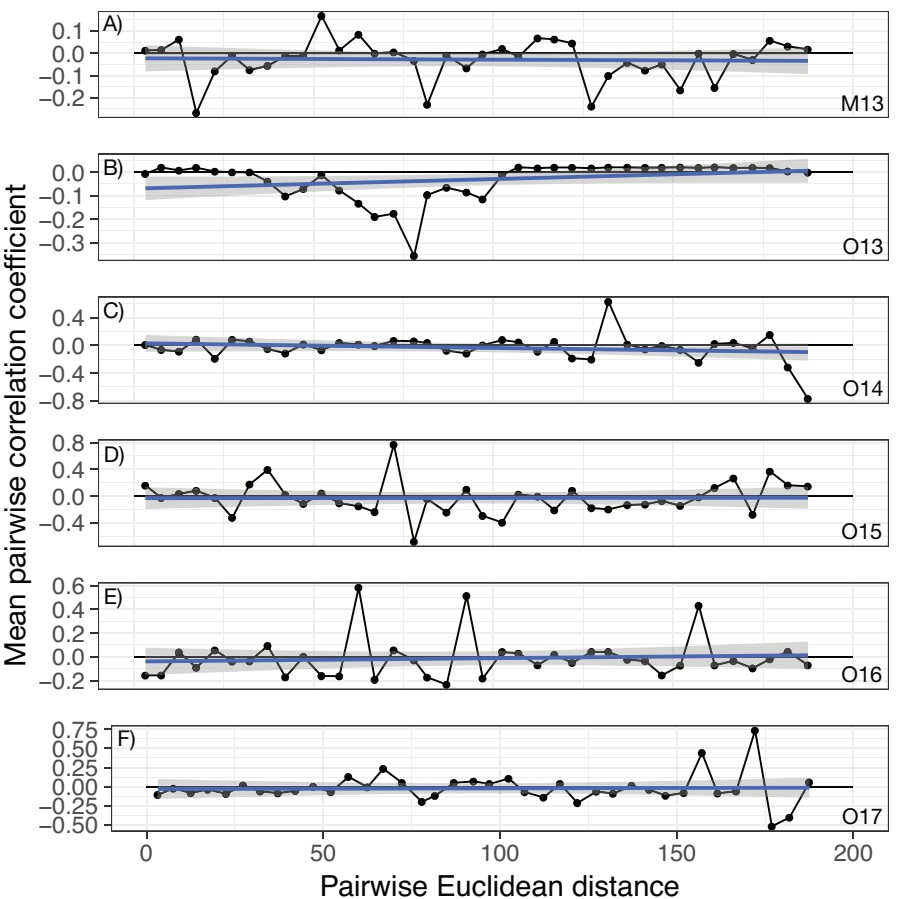

**Figure 1** **Annual correlograms of target isolate abundance by vine distance.** Correlograms for samples taken in (A) May 2013, (B) October 2013, (C) October 2014, (D) October 2015, (E) October 2016, and (F) October 2017. No spatial autocorrelation was detected in any time frame (see Table S1 for statistics), which would appear as consistent, positive correlation at shorter distances. Each point represents the mean correlation coefficient for all pairwise vine combinations within the range of distance represented by that point. The black line represents the zero correlation line, the blue line shows the regression line, and the shading depicts the 95% confidence interval on the regression line.

$n = 104$, $P = 0.38$). When subset by strategy, inoculation did not decrease the mortality rate in any strategy (Fisher's Exact, Pre—odds ratio = 1.081, $n = 33$, $P = 1$; Co—odds ratio = 2.040, $n = 35$, $P = 0.44$; Est—odds ratio = 1.770, $n = 36$, $P = 0.48$).

## Shoot length

Median shoot length during 2013–2015 measured 86.5 cm (range = 14–182 cm) for non-inoculated vines, and 85.25 cm (range = 11–259 cm) for inoculated vines, and was not significantly different between these two groups (Fig. S10; LMM, $X^2 = 0.830$, $n = 270$, $P\text{-} = 0.362$). There was also no interaction between inoculation and strategy (LMM, $X^2 = 2.999$, $n = 270$, $P\text{-} = 0.223$).

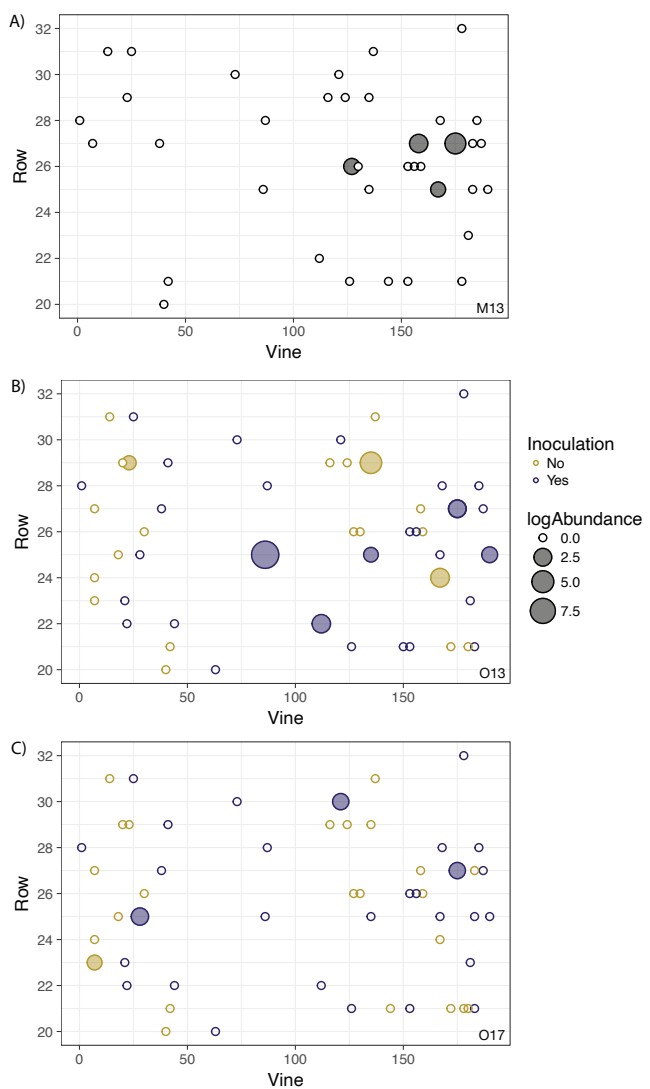

**Figure 2 Geographic distribution and abundance of the target isolate, *R. irregulare* DAOM 197198, in soil samples taken in May 2013 (A) and October 2013 (B), and October 2017 (C).** Each point represents an individual experimental vine, and the point is filled if the sample tested positive for the presence of the isolate or left unfilled if the sample tested negative. The size of the point is proportional to the log abundance of the isolate. Yellow points represent non-inoculated vines, and purple points represent inoculated vines (colours omitted in A due to being pre-inoculation). No statistically significant spatial autocorrelation is present.

## Shoot base diameter

Median shoot diameter was very similar between inoculated (6.2 mm; range 2.9–9.0 mm) and non-inoculated vines (6.0 mm; range 2.4–9.9 mm), and was not significantly different (Fig. S11; LMM, $X^2 = 1.070$, $n = 190$, $P = 0.301$). Again, there was no interaction between inoculation and strategy (LMM, $X^2 = 0.424$, $n = 190$, $P\text{-} = 0.810$).

### Grape cluster production
#### Binary cluster production
Overall, inoculated vines were not more likely to produce grapes (60.8%) compared to non-inoculated vines (69.0%; Fig. S12; Fisher's Exact, odds ratio = 0.700, $n = 88$, $P$- = 0.505). This remained consistent within different strategies (Fisher's Exact; Pre—odds ratio = 1.241, $n = 29$, $P = 1$; Co—odds ratio = 0.440, $n = 32$, $P = 0.442$; Est—odds ratio = 0.636, $n = 27$, $P = 0.636$).

#### Cluster number
Within vines that produced fruit, inoculated vines did not produce more grape clusters (median = 3, range = 1–24) than did non-inoculated vines (median = 3, range = 1–8; Fig. S13; LMM, $X^2 = 0.427$, $n = 70$, $P$- = 0.514). The interaction between inoculation and strategy did not have a significant effect (LMM, $X^2 = 2.720$, $n = 70$, $P$- = 0.257).

## DISCUSSION

Contrary to our expectations, pre-inoculation did not ensure successful establishment of the commercial inoculant in the field. There were few positive samples overall, and inoculated samples were no more likely to be positive than non-inoculated samples. Root and soil samples from interrow vegetation also showed very low abundance of the inoculant. Thus, inoculation of plants with the commercial inoculant was unsuccessful regardless of whether it was conducted pre-transplant in the greenhouse, during transplant in the field, or on one-year-old vines in the field.

Failure of AM fungal inoculants establishment in the field is not unusual. In one study, 9 out of 12 inoculants failed to establish in sterile soil in greenhouse, and 5 of 12 failed to establish in non-sterile soil (*Faye et al., 2013*). Establishment failure has been observed in other greenhouse trials using both sterile and non-sterile soil (*Corkidi et al., 2004*; *Tarbell & Koske, 2007*; *Köhl, Lukasiewicz & Van der Heijden, 2016*), and also in field trials of inoculants (*Berruti, Lumini & Bianciotto, 2017*).

The equal likelihood of detection in both inoculated and non-inoculated samples and detection of the isolate pre-inoculation indicate that positive samples could have been from a pre-existing background population. However, it is not clear whether this is due to deliberate introduction through commercial inoculants simply the natural range of this isolate. The ranges of AM fungal species are very difficult to determine, although some AM fungal species have been found across widespread distributions (*Davison et al., 2015*; *Oehl et al., 2017*; *Savary et al., 2018*). Therefore, the range of *R. irregulare* DAOM 197198 could conceivably extend from its original collection location in Quebec to our site in British Columbia. Nonetheless, we know that this isolate was not introduced in our site within the last 15 years and we were unable to find other records from natural ecosystems in British Columbia.

### Possible causes of establishment failure
The goal of this experiment was to test the influence of priority effects on AM fungal inoculant establishment; however, the inoculant failed to establish in the field in any

treatment. One factor that may have contributed to poor establishment in our experiment is the soil chemistry at the site. Soil phosphorus at the site was roughly ten times more abundant than the 20–50 mg/kg recommended for organic grape production in the region (*Gough, 1996*; *Landers et al., 2011*). If plants are grown under surplus phosphorus conditions, the nutritional benefit of the partnership can be outweighed by the carbon cost (*Johnson, 2010*). Elevated soil phosphorus may inhibit root colonization altogether (*Balzergue et al., 2011*; *Balzergue et al., 2013*; *Nouri et al., 2014*). Median root fungal colonization in our grapevines was 35.5%, which is roughly half of the typical 60–80% root colonization for the rootstock used here (*Schreiner, 2003*).

Additionally, this experiment was conducted in an organic vineyard, which often has higher microbial biomass than conventional viticulture (*Karimi et al., 2020*) and can also increase mycorrhizal root colonization (*De Freitas et al., 2011*). This increased microbial biomass and activity may exacerbate priority effects and contribute to establishment failure in organic systems in general, although in our system root fungal colonization was relatively low and pre-inoculation did not improve inoculation success over inoculation in the field. Additionally, the amount and method of inoculum application can affect the strength of colonization (*Van Jaarsveld et al., 2021*).

Although the roots were trimmed during planting, it has been shown that this practice does not remove all of the active mycorrhiza and colonization can be re-established from mycorrhizas remaining in the older portion of the root system (*Holland et al., 2019*).

## Implications for practical applications of biofertilizers

Successful establishment of AM fungal inoculants is often unreliable, and our results confirm other studies showing high rates of inoculant failure (*Corkidi et al., 2004*; *Tarbell & Koske, 2007*; *Faye et al., 2013*; *Berruti, Lumini & Bianciotto, 2017*; *Köhl, Lukasiewicz & Van der Heijden, 2016*). To assess the value of using an inoculant, accurate isolate-specific tracking is essential, because in this case the lack of growth benefits was due to poor inoculant establishment rather than a well-established but poorly beneficial inoculant. It is possible that *R. irregulare* DAOM 197198 may be effective in producing increased growth, yield, or survival in grapevine if it does establish; however, for growers using these inoculants, uncertainty in the likelihood of establishment remains a significant barrier. Without successful establishment of the isolate, no direct effects of the biofertilizer on the grapevine were possible (although sometimes indirect effects may occur through changes to the soil microbial community, see *Berruti, Lumini & Bianciotto, 2017*), which is ultimately what growers using these inoculants would like to achieve.

In addition to the financial costs to the producers with inconsistent and unreliable return on investment, these inoculants may also present an ecological cost through the risk of invasion (*Schwartz et al., 2006*; *Hart, Antunes & Abbott, 2017*; *Ricciardi et al., 2017*). When this isolate is introduced to an established AM fungal community, it can significantly reduce the species richness by displacing native AM fungal species (*Koch et al., 2011*; *Symanczik et al., 2015*). Aboveground plant communities are closely linked to the identities and diversity of AM fungal communities (*Van der Heijden et al., 1998*;

*Van der Heijden, Wiemken & Sanders, 2003*; *Stampe & Daehler, 2003*; *Bever et al., 2010*), which the introduction of a commercial isolate has the potential to disrupt.

## CONCLUSIONS

In this study, we examined whether a commercial biofertilizer was beneficial in grapevine inoculated in three different strategies; however, the fungus in the biofertilizer did not successfully establish in any treatment. These results highlight the fact that achieving benefits with biofertilizers will require careful site by site evaluation. These products often claim to guarantee results, which is deceptive as it is not enough to simply introduce the isolate and expect changes to plant performance. Many management practices will affect AM fungi both positively and negatively, which should be evaluated prior to use of AM fungal biofertilizers. For example, our results show that it is important for users to know and regularly test their soil fertility, to avoid inoculating over-fertilized soils. Also, other management practices such as reduced tillage and shortened fallow have been shown to be beneficial to AM fungi (*Lekberg & Koide, 2005*). Therefore, while the use of AM fungal inoculants would likely be effective under certain conditions where the colonization potential would otherwise be very low, the potential of these inoculants to mitigate the impacts of declining phosphorus availability should be viewed critically in the light of the costs associated with introduction.

## ACKNOWLEDGEMENTS

Thank you for Kalala Organic Estate Winery for their support.

### Funding

This work was supported by an NSERC Engage Grant and NSERC Discovery Grant to Miranda Hart and an NSERC CGS-M to Corrina Thomsen. The funders had no role in study design, data collection and analysis, decision to publish, or preparation of the manuscript.

### Grant Disclosures

The following grant information was disclosed by the authors:
NSERC Engage Grant.
NSERC Discovery Grant.
NSERC CGS-M.

### Competing Interests

The authors declare there are no competing interests.

### Author Contributions

- Corrina Thomsen performed the experiments, analyzed the data, prepared figures and/or tables, authored or reviewed drafts of the paper, and approved the final draft.

- Laura Loverock and Vasilis Kokkoris performed the experiments, authored or reviewed drafts of the paper, and approved the final draft.
- Taylor Holland performed the experiments, authored or reviewed drafts of the paper, and approved the final draft.
- Patricia A. Bowen and Miranda Hart conceived and designed the experiments, authored or reviewed drafts of the paper, and approved the final draft.

## Field Study Permissions

The following information was supplied relating to field study approvals (i.e., approving body and any reference numbers):

Field site permission was provided by Ashish Dave, Research Manager Kalala Estate Winery, West Kelowna, BC and Karnail Sidhu, Owner Kalala Estate Winery, West Kelowna, BC.

## Data Availability

Raw data are available in the Supplemental Files.

## Supplemental Information

Supplemental information for this article can be found online at http://dx.doi.org/10.7717/peerj.11119#supplemental-information.

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
