# Peer review of "Commercial arbuscular mycorrhizal fungal inoculant failed to establish in a vineyard despite priority advantage"

_PeerJ, doi:10.7717/peerj.11119_

## Round 0.1 · original submission · Major Revisions

Dear Dr. Hart,

Thank you for your submission to PeerJ.

It is my opinion as the Academic Editor for your article - Commercial arbuscular mycorrhizal fungal inoculant failed to establish in a vineyard despite priority advantage - that it requires a number of Major Revisions.

My suggested changes and reviewer comments are shown below and on your article 'Overview' screen.

Please address these changes and resubmit. Although not a hard deadline please try to submit your revision within the next 55 days.

Reviewer 1 ·

Basic reporting

In this article, the authors evaluated the performance of a commercial AMF product to be established in vineyards. Although the experiment is very simple, it is novel in terms of the potential applicability to the industry. It shows the reality of this type of products. Similar experiments with more biocontrol agents should be performed in the near future. My major criticism is that the authors only used one grapevine rootstock genotype and the experiment was performed in only one vineyard plot. Rootstock genotype and soils with different physico-chemical properties should had been included in this study.

Apart of these comments the manuscript is well written and will significantly contribute to add to our current knowledge on AMF and thus, I recommend publication after revision. Literature references are sufficient and the structure of the paper is appropriate. Hypotheses and results are well connected.

Experimental design

Regarding the experimental design, it is not clear to me how many vines were used per treatment. Perhaps it should be described better in the text instead of doing it in the Supp material. Apart from this comments, methods are described with sufficient detail and information to replicate.

Validity of the findings

In general, all relevant data has been showed in the manuscript. Conclusions are well stated and linked to the original research questions.

I have only a comment regarding the ddPCR positive results in Lines 311-314 & Table 2. Bio-Rad’s Droplet Digital PCR Applications Guide says that if a single wild-type or negative sample control well is run and the observed positive droplets are zero, then it is good practice to require at least three positive droplets in order to call a sample positive, and the three positive droplets can be in either a single well or across merged wells. I think the authors should follow the most restrictive recommendation of Bio-Rad's guidelines in order to avoid potential false positives to be account as positive samples for the quantification of the AMF from the interrow soil and vegetation roots samples, in order to have better accurate data.

Additional comments

In this article, the authors evaluated the performance of a commercial AMF product to be established in vineyards. Although the experiment is very simple, it is novel in terms of the potential applicability to the industry. It shows the reality of this type of products. Similar experiments with more biocontrol agents should be performed in the near future. My major criticism is that the authors only used one grapevine rootstock genotype and the experiment was performed in only one vineyard plot. Rootstock genotype and soils with different physico-chemical properties should had been included in this study.

Apart of these comments the manuscript is well written and will significantly contribute to add to our current knowledge on AMF and thus, I recommend publication after minor revision.

Reviewer 2 ·

Basic reporting

English is clear and professional.
The literature and background provided are good. One key author is missing in the discussion, the work from Lionel Ranjard. For example Karimi, B., Cahurel, JY., Gontier, L. et al. A meta-analysis of the ecotoxicological impact of viticultural practices on soil biodiversity. Environ Chem Lett 18, 1947–1966 (2020). https://doi.org/10.1007/s10311-020-01050-5
Please add the importance of the natural soil communities (plant-associated or not) on the efficient colonization of BCA or AMF. Organic production seems to favour soil microbial diversity. This would mean the higher the diversity, the higher the chances to get low colonization by the BCA of AMF because the ecological niche is already taken.
The application method is also important, like presented recently in Trichoderma work by Wynand J van Jaarsveld et al. 2020. Investigation of Trichoderma species colonization of nursery grapevines for improved management of black foot disease. https://doi.org/10.1002/ps.6030, which may enrich the discussion.

Experimental design

The field trial is well conducted.
However the reader needs more details about the product tested. Your titles generalized the results to all preparations possible, but it focuses on a product actually advertised for green house usage. How does the company advices to inoculate the product when planting in soil?

Is there a link between those strains: Rhizophagus irregularis DAOM 181602=DAOM 197198 because I found this accession when looking for sequences on ncbi ?
I looked at the product MYKE® PRO GREENHOUSE • G on manufacturer’s website. The active ingredient is Glomus intraradices, current name being Rhizophagus. Please clarify and cite the description by Schuessler, A. and Walker, C. 2010. The Glomeromycota: a species list with new families and new genera 1-58. electronic publication. Available at: www.amf-phylogeny.com to clarify this.

Validity of the findings

This is the critical point for this manuscript. It is difficult to prove that something is not present. In order to strengthen this work, authors should solidify their molecular diagnosis. They are few positives, and after sequencing they do not particularly match the reference sequences on ncbi. It is difficult to be sure, reading at the data presented, that the molecular tool actually worked in these conditions.
I would like to see a basic PCR test from a pure culture of R. irregulare showing an expected product, with a confirmation by sequencing. In addition it is disturbing to read on the website company that the active ingredient seems to be another species within the same genus, Glomus intraradices.

Please do not generalize too much this negative result to all the AMF product because only one was tested, in one location of one combination cultivar x rootstock.

Additional comments

Dear Editor, the manuscript “Commercial arbuscular mycorrhizal fungal inoculant failed to establish in a vineyard despite priority advantage” submitted by Corinna Thomson and colleagues’ addresses a key question in agroecology: inoculation of BCA/AMF and their persistence in the agroecosystem. The manuscript is well written and the field trial well conducted. Despite author’s expectation the commercial product used did not colonized the grafted plants, with a very low number of positive. I recognize the important amount of work realized to test all the samples. I agree with authors that it is important to publish this result, but it is difficult to prove that a phenomenon is not in science. To do so the molecular biology needs to be more convincing and confusion clarified.
Indeed the few positives has been send for sequencing, and sequences do not match the reference one in ncbi. A basic PCR showing that the primers actually amplify the fungus in from pure culture and from the control artificially inoculated would help, with a confirmation of the sequences. I guess the artificially inoculated plant use as positive showed amplification in the ddPCR analyses, confirming the sequence would strengthen the manuscript. Finally, because few details are given about the commercial product and its uses, I went on the website of the manufacturer. This product is design or advertised for greenhouse usage, but more importantly the active ingredient is Glomus intraradices which seems a relative but different species that the one targeted in the molecular diagnostic.

For all those reasons I would recommend to reject/major revisions the manuscript and invite authors to resubmit after clarifying these points. I encourage authors to do so because these experiments are really important for helping vinedressers to move to sustainable farming, and unfortunately they are still too rare.

---

## Round 0.2 · accepted · Accept

No additional comment from my side.

Reviewer 1 ·

Basic reporting

No comment

Experimental design

No comment

Validity of the findings

No comment

Additional comments

The article is good in its current form

Reviewer 2 ·

Basic reporting

The article completes all those criteria

Experimental design

- Scope of the journal: negative result important to share for improving the development of efficient AMF/BCA agents.
- Research question relevant and well defined
- Rigorous investigation with high technical and ethical standards
- Method described with sufficient detail: thank you for the nice explanation, in supplement data, of the ddPCR molecular technique and the way you handle the data, fix your thresholds.

Validity of the findings

- Negative result well described and relevant for our research community

Additional comments

The manuscript entitled “Commercial arbuscular mycorrhizal fungal inoculant failed to establish in a vineyard despite priority advantage” authored by Corrina N. Thomsen and colleagues investigates the establishment of a AMF product in an organic wineyard five years after treatment.
This study is very precious because colonization success is a key question in developing new BCA or biofertilizers for a sustanaible viticulture. Despite the negative result, the work is well conducted and pleasant to read. Very interesting description of the way to handle and analyse ddPCR data.
All the question I had during the first round of rewieving has been answered clearly and I thank authors for this.
I recommend to accept this manuscript for publication.